# A Self-Bleaching Electrochromic Mirror Based on Metal Organic Frameworks

**DOI:** 10.3390/ma14112771

**Published:** 2021-05-24

**Authors:** Kun Wang, Kai Tao, Ran Jiang, Hongliang Zhang, Lingyan Liang, Junhua Gao, Hongtao Cao

**Affiliations:** 1Institute of Inorganic Materials, School of Materials Science and Chemical Engineering, Ningbo University, Ningbo 315211, China; wangkun1995@nimte.ac.cn (K.W.); taokai@nbu.edu.cn (K.T.); jiangran@nbu.edu.cn (R.J.); 2Laboratory of Advanced Nano Materials and Devices, Ningbo Institute of Materials Technology and Engineering, Chinese Academy of Sciences, Ningbo 315201, China; lly@nimte.ac.cn (L.L.); gaojunhua@nimte.ac.cn (J.G.)

**Keywords:** MOFs, WO_3_, electrochromic mirror, self-bleaching, reflectance modulation

## Abstract

Metal-organic frameworks (MOFs) are considered to be the most promising positive anode materials to store charge for electrochromic devices. Nevertheless, a detailed mechanism of the electrochemical and ions storage process has not yet been revealed. Herein, the electrochemical mechanism of the highly porous ZIF-67 films and the electrochromic performance of electrochromic mirrors constructed from ZIF-67 and WO_3_ electrodes were investigated. The mechanism of the charge storage was revealed in the kinetic analysis of the Li-ion behavior based on the cyclic voltammetry curves and electrochemical impedance spectra. Impressively, the electrochromic mirrors with the self-bleaching effect and self-discharge behavior showed a unique electrochromic performance, such as a high coloration efficiency of 16.47 cm^2^ C^−1^ and a maximum reflectance modulation of 30.10% at 650 nm. This work provides a fundamental understanding of MOFs for applications in electrochromic devices and can also promote the exploration of novel electrode materials for high-performance reflective electrochromic devices.

## 1. Introduction

Electrochromic devices (ECDs) have received widespread attention due to their wide range of applications, including antiglare rearview mirrors [1], smart windows [2,3] and information displays [4,5]. A typical electrochromic (EC) device generally consists of five superimposed layers, an ion electrolyte layer sandwiched between an EC layer and an ion storage layer (counter electrode) that are individually deposited on transparent conductive electrodes [6]. For instance, tungsten oxide (WO_3_), one of the most typical EC layers, which changes its optical properties reversibly (absorbance/transmittance/reflectance) via redox reactions under an alternating potential or current modulation [7], has been considered as the most promising cathodic electrochromic candidates for EC applications owing to its excellent EC properties, such as high cyclic stability and coloration efficiency [8,9]. In view of the requirement of charge matching and redox balance, another complementary anodic EC ion storage layer based on WO_3_ is commonly used as an indispensable component in ECDs [10]. Conventionally, a complementary electrochromic device is composed by assembling both WO_3_ and ion storage layers, such as NiO [11], CeO_2_ [12] and Prussian blue (iron ferrocyanide) analogues [13], which color under the insertion or extraction of small ions, respectively. Developing ion storage layer materials is an interesting scientific challenge with promising applications in electrochromic devices. Very recently, many researchers have focused their attention on novel multifunctional MOFs [14], supramolecular assemblies formed by the reaction of metal nodes with organic linkers due to the unique features, such as high specific surface area, large pore size and tunable channels for ionic diffusion [15,16,17]. An interesting feature of MOFs is their ability to allow the rapid insertion/extraction of ions and, consequently, offer the possibility to improve their electrochemical performance [18,19]. For instance, a 2-dimensional layered nickel-based MOF has a high specific capacity of 320 mA h g^−^^1^ for the storage of Li^+^ [20]. Highly uniform graphene shell coated 1D NiCo_2_O_4_ electrodes, prepared by adjusting the reaction component of ZIF-67, have been reported to enjoy a superior ion storage capacity and cycling stability [21]. Li Z. et al. have demonstrated that the thin-layer structural ZIF-67 electrodes, prepared using a drop-casting method, exhibit a high reversible capacity of 311.6 mA h g^−^^1^ and a good cycling stability [22]. Therefore, it is worth exploring the promising ZIF-67 anode materials as an ion storage layer to satisfy the future applications of high performance WO_3_-based ECDs.

Herein, a novel electrochromic mirror based on ZIF-67 MOFs and WO_3_ films was constructed. The investigation into the configuration and working mechanism of the electrochromic devices on a basis of Li^+^-containing electrolyte was performed. A schematic diagram of the mirror mode operation principle is depicted in Figure 1. The obtained reflection-type ECDs with a self-bleaching effect possess a high coloration efficiency of 16.47 cm^2^ C^−^^1^.

## 2. Materials and Methods

### 2.1. Materials

All of the reagents used were of an analytical grade. Cobalt nitrate hexahydrate (Co(NO_3_)_2_·6H_2_O) and absolute ethyl alcohol (ethanol, analytical grade) were purchased from Aladdin Chemical (Shanghai, China) and Sinopharm Co. Ltd. (Beijing, China), respectively. The conducting transparent indium tin oxide (ITO)-coated glasses were used as the substrate of the WO_3_ and Ag thin films deposited using the electron beam evaporation technology.

### 2.2. Synthesis

#### 2.2.1. Preparation of the MOF/Ag films

The approximately 150-nanometer-thick Ag thin films were deposited using the electron beam evaporation technique (MUE-ECO made in ULVAC, Chigasaki, Japan) at a deposition rate of about 0.10 nm/s and a background pressure of less than 2.00 × 10^−3^ Pa. Subsequently, the MOF films were electrochemically deposited onto the ITO substrates with Ag thin films from a solution containing 40 mL of ethanol, 0.8 M of 2-methylimidazole and 0.1 M of Co(NO_3_)_2_·6H_2_O in a conventional three electrode cell. The Pt sheet, ITO substrate with Ag thin films and SCE were used respectively as counter, working and reference electrodes. The MOF films were obtained by applying a constant voltage of −5 V to the ITO substrates for 10 mins. The electrodeposited films were cleaned using ethanol and deionized water. Finally, the films were dried at 60 °C under air atmosphere for 6 h.

#### 2.2.2. Preparation of the WO_3_ thin Films with SnO_2_ Interface

The WO_3_ thin films were deposited onto the ITO-coated glass with a deposition rate of 0.10 nm/s and a thickness of about 450 nm, at a substrate temperature of 200 °C in a vacuum of less than 2.0 × 10^−^^3^ Pa, using an electron beam evaporation technique (MUE-ECO made in ULVAC, Chigasaki, Japan). Subsequently, the SnO_2_ ultra-thin interfacial layers (thickness: ~5 nm) were grown in situ on the WO_3_ thin films using the same technique.

#### 2.2.3. Assembly of the Mirrors

The electrochromic mirrors, with an ITO/WO**_3_**/SnO**_2_**/electrolyte/ZIF-67/Ag/ITO configuration, were assembled by filling an 0.10 M PC-LiClO**_4_** electrolyte into the SnO**_2_**/WO**_3_** thin film and the MOF/Ag film in the vacuum, as reported previously [23].

### 2.3. Measurements

The structure and morphology of the MOF/Ag films were determined using an X-ray diffraction (XRD, D8 Advance, Bruker, Germany) with Cu-Kα radiation (λ = 0.154178 nm), a thermal field-emission scanning electron microscopy (TFESEM, Sirion200, FEI, Hillsboro, OR, USA) and an X-ray photoelectron spectra (XPS) (AXIS UTLTRA DLD, Kratos, Manchester, England). Fourier-transform infrared (FTIR) transmission spectra were obtained using a micro-FTIR spectroscopy (μ-FTIR, Cary660+620, agilent, Santa Clara, CA, USA). In situ optical reflectance spectra were obtained using UV–VIS–IR spectroscopy (Lambda 950, Perkin-Elmer, Waltham, MA, USA) and an electrochemical workstation (CHI660D, Chen hua, Shanghai, China). The cyclic voltammetry measurements were conducted by applying a voltage between 0.00 V and +0.60 V to MOF/Ag films in a three-electrode cell, which employs a platinum sheet, KCl-saturated Hg/HgCl_2_ and 0.10 M LiClO**_4_**-PC electrolyte as a counter electrode, a reference electrode and an electrolyte, respectively. The electrochemical impedance spectra were carried out using an electrochemical workstation (Zennium, IM6) in a frequency range of 10^−^^1^–10^5^ Hz. The OCV and cyclic voltammetry measurements of the ECDs were carried out in a two-electrode cell.

## 3. Results and Discussion

### 3.1. Characterization of Films

Figure 2a shows XRD patterns of the ZIF-67 powder, ZIF-67/Ag/ITO, Ag/ITO and bared ITO. As can be seen, the peaks located at 31°, 37° and 51° are assigned to In_2_O_3_, indium and tin [24]. For the Ag/ITO, several characteristic peaks at 38°, 44° and 64.5° can be observed, corresponding respectively to (111), (200) and (220) planes [25]. This reveals that a single phase with a cubic geometry and space group Fm-3m (225) for Ag is formed. The peaks at 7.40°, 10.35° and 18.03° are respectively indexed to the crystalline ZIF-67 phase (011), (002) and (222) [26], which indicates that the electrodeposited ZIF-67 films enjoy a crystalline characteristic. The average size of the ZIF-67 film is calculated to be about 26.31 nm from the XRD data using Scherrer’s formula. Besides, the two diffraction peaks at 21.01° (114) and 25.32° (233) found in the powder sample are absent in the film sample. The highly porous morphology can be clearly observed in the SEM image in Figure 2b, which is conducive to the insertion of ions. In addition, the elemental compositions of ZIF-67 films can be determined using EDS analysis, as shown in the inset of Figure 2b. Figure 2c shows the FTIR spectra of the ZIF-67 films and powder. In Figure 2c, the absorption peak at 1575 cm^−1^ is assigned to the C=N band. There is a strong stretching band of O-H at about 3200 cm^−1^. In addition, blending vibrations of alkane C-H are observed at 1450 cm^−1^. The peak positions of the film are corresponded basically with that of the powder and are consistent with a previous report [27], which confirms that the film is composed of the ZIF-67 MOFs. In order to further demonstrate the composition details of the resulting samples, an XPS was performed. The high-resolution XPS spectra of Co 2p is shown in Figure 2d. The two major peaks with binding energies of 780.61 eV and 796.17 eV are respectively assigned to Co 2p_3/2_ and Co 2p_1/2_, which correspond to the intrinsic peaks for ZIF-67. The other two indistinctive peaks at 785.00 and 801.41 eV are satellite peaks of Co [28]. The results show that the two valence states (Co^2+^ and Co^+^) coexist, similar to a previous report [29].

### 3.2. Electrochemical Properties of the ZIF-67/Ag Films

The cyclic voltammetry curves of ZIF-67/Ag electrodes at different scan rates, from 5 to 10 mV/s between 0.00 and +0.60 V, are shown in Figure 3a. It can be observed that the peak value of the current density grows accordingly with the increased scan rate. There is no obvious difference for the peak current density between 5 and 6 mV/s, probably due to the saturation of the ion diffusion. It is worth noting that a small shift in the reduction peak with an increasing scan rate can be observed, which indicates the excellent stability of the reaction on the electrode active surface, consistent with a previous report [30]. Moreover, the faradaic and capacitive-controlled processes are identified by the ‘b’ value, which is calculated using the following equation [31]:i = av^b^(1)
where i and v represent, respectively, the peak current and the scan rate. Both ‘a’ and ‘b’ are the adjustable parameters. It is noteworthy that the ‘b’ value determined from the slope of log(i) versus log(v) is close to 0.5 and 1.0, indicating that the current response belongs predominantly to the diffusion controlled and the capacitive processes [32], respectively. As shown in Figure 3b, the ‘b’ value of the ZIF-67/Ag films is calculated to be 0.59 (between 0.5 and 1), indicating that the charge storage is composed of diffusive charges and capacitive charges. Besides, in order to further understand the charge transfer and ion diffusion processes of the ZIF-67/Ag films, electrochemical impedance spectra tests were performed. Figure 3c displays the Nyquist plots measured from the ZIF-67/Ag films with +0.60 V and 0.00 V applied the Li^+^ electrolyte in the frequency region (100 mHz to 100 kHz). The impedance spectra can be evaluated using the equivalent electrical circuit given in the inset of Figure 3c. The resistance (R_s_) of the electrolyte, the interfacial charge-transfer resistance (R_ct_) and W correspond to the intercept of the real axis in the high frequency region, the diameter of the semicircle and the semi-infinite Warburg element, respectively [33]. The fitted data (Table 1) reveals that the R_ct_ of the ZIF-67/Ag films is respectively estimated to be approximately 82.81 Ω cm^−2^ and 6.75 Ω cm^−2^ when 0.00 V and +0.60 V are applied, indicating that the ZIF-67/Ag films at +0.60 V possess a higher conductivity, as stated in a previous report [34]. In addition, the intercalation/extraction capacity of electrodes is a reciprocal function of the absolute value of the imagine resistance at a low frequency [35]. Thus, it can be seen that the extraction of ions is prominent at 0.6 V. Moreover, the relation between the frequency and the phase angle is shown in Figure 3d. In the lower-frequency-limitation region, the phase angle (−62.12°) of +0.60 V is significantly higher than that of 0.00 V (−19.96°), indicating a faster ion diffusion in the electrolyte, in agreement with a previous report [36].

### 3.3. Electrochromic Performance of the WO_3_//ZIF-67 Mirrors

Figure 4a displays the in situ reflectance spectra for the ECDs at 650 nm, obtained by alternately applying a −1.00 V and 0.00 V voltage both for 40 s and 100 s, respectively. The maximum reflectance modulation (△R, coloration/bleaching at 650 nm) is estimated to be 30.10%. The coloring and bleaching response time (defined as the time required to reach 90% of the reflectance change) is calculated to be 33 s and 30 s, respectively. The lower bleaching time can be attributed to a reduction in the interfacial charge-transfer resistance, as analyzed using the impedance spectra. It is worth noting that no significant change in the regulation of reflectivity of the reflection-type electrochromic devices after nine cycles is observed, indicating a relatively stable optical modulation. In Figure 4b, the devices need 100 s and 300 s, respectively, to reach the peak reflectivity, when a voltage of 0.0 V is applied for 100 s and removed after a voltage of −1.0 V is applied for 40 s. These results may be due to the strong adsorption of ions by the MOF [37], which leads to the spontaneous deintercalation of Li ions at the interface of the WO_3_/electrolyte, resulting in a corresponding self-bleaching effect. Interestingly, the WO_3_//ZIF-67 mirrors without ZIF-67 seem to maintain the initial low reflection mode more easily after the voltage is removed, in comparison to the WO_3_//ZIF-67 mirrors with ZIF-67, as shown in the in situ reflectance spectra in Figure 4c. This result further confirms that the self-bleaching effect is caused by ZIF-67. The coloration efficiency (CE), as one of the most important characteristics of ECDs, represents the ability to achieve optical modulations with changes in energy or charge. The CE, which is defined as the change in optical density per injected charge density at a particular wavelength, is calculated to be 16.47 cm^2^ C*^−^*^1^ (Figure 4d), larger than the 12.60 cm^2^ C*^−^*^1^ reported previously [38]. The improved electroactivity reflected by the encapsulated area of the CV curve can be seen after 100 cycles (Figure 4e), suggesting a good electrochemical stability of the mirrors, in line with a previous report [39]. Figure 4f exhibits the OCV experimental results of WO_3_//ZIF-67 devices. It can be observed obviously that the OCV of the ECDs declines rapidly within 400 s and the decay of the OCV becomes slower after 400 s, which corresponds to the self-bleaching behavior.

## 4. Conclusions

In summary, the electrochemical properties of the electrochemically deposited ZIF-67 films in the Li^+^-based electrolyte have been analyzed. Both the semi-infinite diffusion and surface-controlled process contributed to the electrochemical reaction of the highly porous ZIF-67 films. The novel electrochromic mirrors constructed from ZIF-67 and WO_3_ electrodes exhibit a high coloration efficiency of 16.47 cm^2^ C^−1^. The self-bleaching effect that originated from the strong adsorption of highly porous ZIF-67 films for ions is demonstrated. The results suggest that the MOF-based electrode materials as an ion storage layer is a promising strategy for promoting the development of reflection-type electrochromic devices.

## Figures and Tables

**Figure 1 materials-14-02771-f001:**
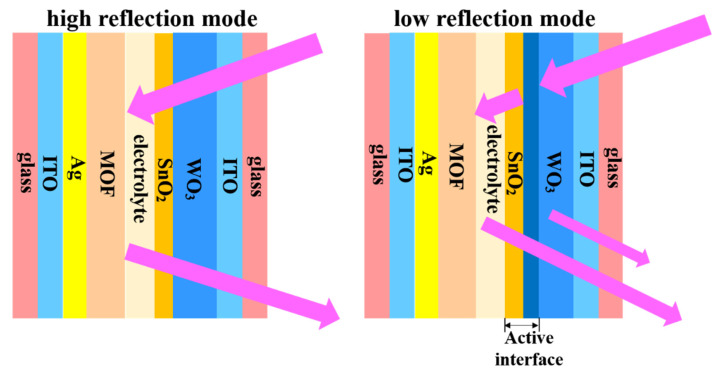
Schematic illustration of the optical modulation mechanism of the WO_3_//ZIF-67 ECD.

**Figure 2 materials-14-02771-f002:**
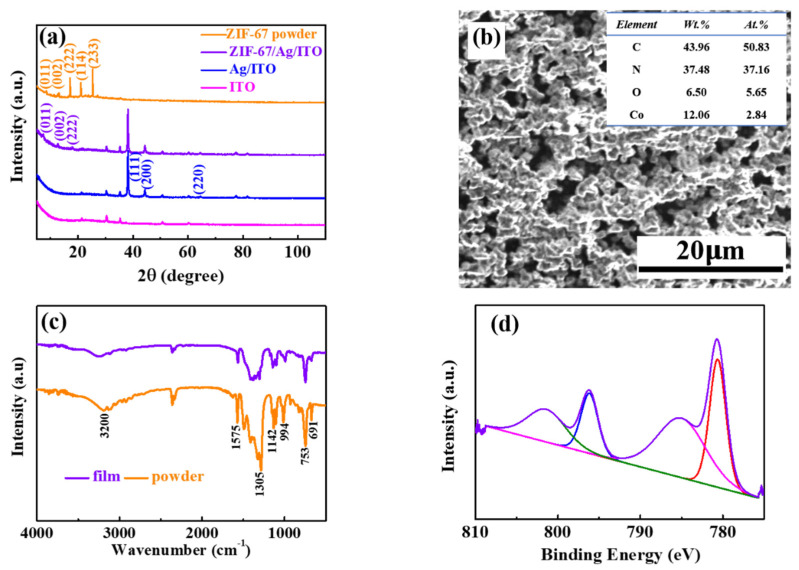
(**a**) XRD patterns of ZIF-67 powder, ZIF-67 films on an ITO substrate with a Ag film, Ag thin films on an ITO substrate and bared ITO. (**b**) SEM image of the ZIF-67 films on an ITO substrate with a Ag film. (**c**) FTIR spectra of ZIF-67 films and powder. (**d**) XPS spectra of Co 2p at the initial state.

**Figure 3 materials-14-02771-f003:**
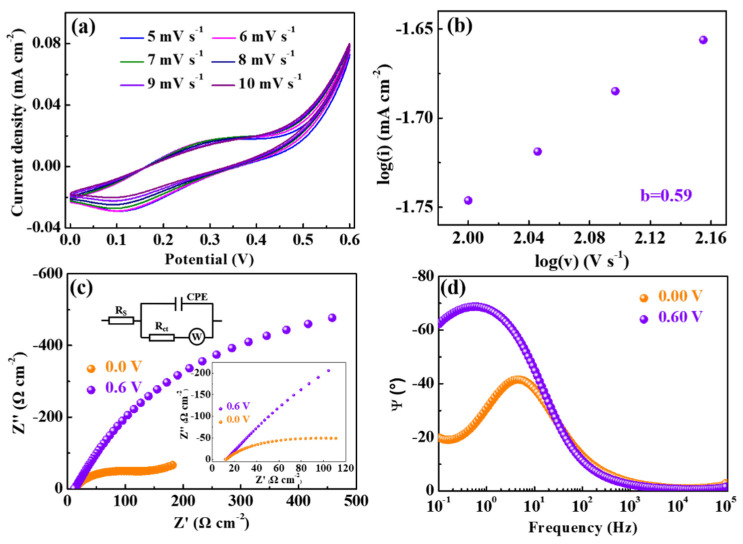
(**a**) CV profiles at various scan rates, (**b**) the power law dependence of the peak current versus the scan rate, (**c**) Nyquist plots and (**d**) the impedance phase angle versus the frequency for the ZIF-67/Ag films in 0.10 M PC-LiClO_4_.

**Figure 4 materials-14-02771-f004:**
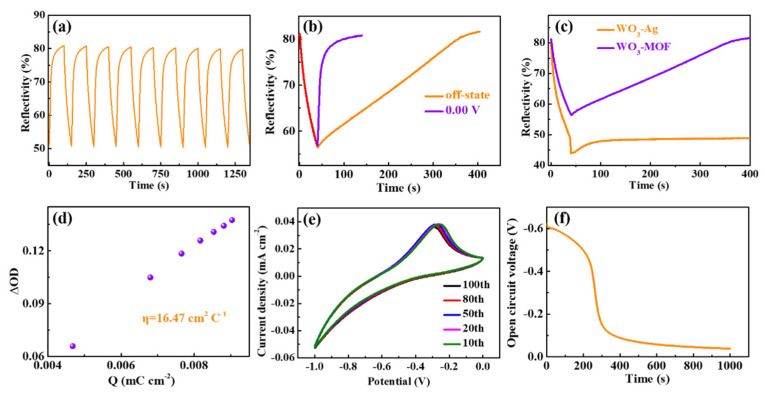
(**a**) In situ time-dependent optical reflectance spectra at λ_650 nm_ (−1.00 V/0.00 V, 140 s per cycle) of the WO_3_//ZIF-67 ECD. (**b**) In situ time-dependent optical reflectance spectra for WO_3_//ZIF-67 ECD when a voltage of 0.0 V was applied for 100 s and removed after a voltage of −1.0 V was applied for 40 s. (**c**) In situ time-dependent optical reflectance spectra for the mirrors with (ITO/WO_3_/SnO_2_/electrolyte/ZIF-67/Ag/ITO) and without (ITO/WO_3_/SnO_2_/electrolyte/Ag/ITO) ZIF-67. (**d**) The plots of in situ optical density variation as a function of charge density at λ_650 nm_, (**e**) cyclic voltammograms with a potential range from −1.00 to 0.00 V at 50 mV s*^−^*^1^ and (**f**) OCV curves for the WO_3_//ZIF-67 ECD.

**Table 1 materials-14-02771-t001:** A summary of the Nyquist measurements for the ZIF-67/Ag films in 0.10 M of PC-LiClO*_4_*.

Applied Potentials (V)	Electrical Parameters	
	R_s_ (Ω/cm^2^)	R_ct_ (Ω/cm^2^)
+0.0	15.07	82.81
+0.6	14.47	6.75

## Data Availability

Data available in a publicly accessible repository.

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
