# Peer review of "A Self-Bleaching Electrochromic Mirror Based on Metal Organic Frameworks"

_materials, 2021, doi:10.3390/ma14112771_

Round 1
Reviewer 1 Report
It is a very nice piece of work by authors. The manuscript can be ready for publishing after going over some minor points below.
1- The manuscript needs a language check for spelling and grammar mistakes. A quick one:" The results suggest is a promising strategy for promoting the development of reflection-type electrochromic devices" in the concl.
2- Figure 2 needs more attention. Its caption should be more descriptive. 2a (pxrd) should have some guiding lines or windows to show related ZIF-67 peaks. Similarly, the oxidation states of cobalt atoms are suggested to show in the XPS figure. DRIFTS figure should highlight the relevant peaks as described in the text. DRIFTS spectra of the bulk and the film MOF should be stacked if possible.
3- please use abbreviations coherently eg. [Co(NO3)2·6H2O] vs Co(NO3)2·6H2O. Just wondering, is there any specific reason that a cobalt mof was prepared over others with different metal nodes?
4-SEM EDS analysis of the MOF layer is recommended. it can give you an idea about the structural homogeneity of ZIF-67 and quantitative analysis for such as Co and Li.
5- Also, any comments on the average sizes of the ZIF-67 crystals and the thickness of the MOF layer?
6-If there is any chance to take krypton sorption, it can reveal the porosity properties of the MOF layer.
7. Equation 1 may be written with the superscript letter 'b' and the comma at the end may be removed.
8. Reference titles should be in the same format.
Best regards,
Reviewer 2 Report
The authors present a novel electrochromic mirrors based on ZIF-67 MOFs and WO3 films. The investigation on the configuration and working mechanism of electrochromic devices on a basis of Li+ containing electrolyte is performed. The presented results seem to imply a promising strategy
for promoting the development of the reflection-type electrochromic devices.
I suggest some minor modifications to be included:
1) Please put hkl index in Figure 2(a), which is mentioned in the text.
2) Page 4, Figure 3 and page 4, lines 49-62. Regarding Nyquist plot and proposed model EC.
3) Could you please put inset for high-frequency data and provide model fits on Figure 3(c).
4) Moreover, value for R ct, charge-transfer resistance. It is not visible from the Figure that for 0 V it is higher than for 0.6 V. Probably it will be better visible when zooming on the high-frequency region. In that case, the dominance of Warburg for 0.6 V measurements can be seen. Please comment. Also, the suggestion is to provide fitting parameters in the additional tables so the comparison will be easier.
Reviewer 3 Report
The manuscript submitted to Materials entitled " A Self-bleaching Electrochromic Mirror Based on Metal-Organic Frameworks" by Kun Wang et al. presents the preparation and electrochromic performance of mirrors constructed from ZIF-67 and WO3 electrodes.
The overall subject has a growing interest. Furthermore, the manuscript's organisation seems logical, and the rationale behind the preparation of electrochemical studies seems logical and, in most cases, well supported by the results. However, in Figure 2, the DRX-powder diffractometer of ZIF-67 should be introduced for comparison with ZIF-67/Ag/ITO, Ag/ITO and bared ITO.
In the same Figure, the authors show the FTIR spectra of ZIF-67 alone. ZIF-67 is one of the most studied MOFs throughout the literature, and this information seems irrelevant without any comparison.
In addition, the authors should compare these results with similar studies from the literature.
